# Effectiveness of a telenursing intervention program in reducing exacerbations in patients with chronic respiratory failure receiving noninvasive positive pressure ventilation: A randomized controlled trial

**Makoto Shimoyama** [1]*, **Shiori Yoshida**[2], **Chikako Takahashi**[3], **Mizue Inoue**[4], **Naoko Sato**[4], **Fumiko Sato**[4]

**1** School of Nursing, Miyagi University, Miyagi, Japan, **2** Department of Health Sciences, Graduate School of Medicine, Tohoku University, Miyagi, Japan, **3** Nursing department, Tohoku University Hospital, Miyagi, Japan, **4** School of Nursing, Fukushima Medical University, Fukushima, Japan

* shimo.ma@myu.ac.jp

## Abstract

Telenursing for patients with chronic respiratory failure receiving noninvasive positive pressure ventilation (NPPV) is an important aid in reducing exacerbations; however, there is insufficient evidence. This randomized controlled trial investigated the effectiveness of a telenursing intervention program in reducing exacerbations in patients with chronic respiratory failure receiving NPPV at home. We included patients receiving NPPV at home who could handle a tablet device. The intervention group (n = 15) underwent an information and communications technology-based telenursing intervention program in addition to usual care; the control group (n = 16) received the usual care only. The telenursing intervention program comprised telemonitoring and health counseling sessions via videophone. The intervention was evaluated once at enrollment and after 3 months. The primary endpoints were the number of unscheduled outpatient visits, hospitalizations, and hospital days. The secondary endpoints included the St. George's Respiratory Questionnaire (SGRQ) score, Euro QOL 5 Dimension score, Self-Care Agency Questionnaire (SCAQ) score, pulmonary function tests, and 6-min walking distance. We used the Mann–Whitney U test for our analysis. We found no significant differences between the intervention and control groups at enrollment. Then, the differences between the endpoints at baseline and 3 months after enrollment were calculated and used to compare both groups. At follow-up, the number of routine outpatient visits for acute exacerbations (p = .045), the number of hospitalizations (p = .037), the number of hospital days (p = .031), SGRQ (p = .039) score, and SCAQ (p = .034) score were significantly different. The increase in the number of unscheduled outpatient visits in the intervention group during follow-up was attributed to acute exacerbations and a significant decrease in the number of hospitalizations and hospital days. Hence, the telenursing intervention program may be effective in reducing exacerbations in patients with chronic respiratory failure receiving NPPV at home.

**Trial registration:** UMIN Clinical Trials Registry (UMIN-CTR) UMIN000027657.

**Data Availability Statement:** All relevant data are within the paper and its Supporting Information files.

**Funding:** The author(s) received no specific funding for this work.

**Competing interests:** The authors have declared that no competing interests exist.

## Introduction

In recent years, as Japan's aging population has been increasing more rapidly than any other country, there has been a call for the establishment of a comprehensive regional care system [1, 2]. A means to support the operation of a comprehensive community care system is telemedicine using information and communication technology (ICT) [3]. In Europe and the United States, telemedicine has become remarkably widespread and has been effective in reducing the cost of home health care by providing the same medical and nursing care to patients in their residences [4, 5]. Telemedicine includes telenursing, in which nurses provide health consultations with patients [6–8] and aims to improve patients' health. Telenursing collects biological information and provides opportunities for accurate health consultation and guidance [9]. Telenursing is gradually spreading in Japan. The International Council of Nurses defines telenursing as the use of telecommunications technology in nursing to enhance patient care [10]. To date, western countries have provided telenursing to patients with chronic obstructive pulmonary disease (COPD) and chronic heart failure [11–13].

Chronic respiratory failure is a condition, in which respiratory failure persists for more than 1 month [14]. Many patients with chronic respiratory failure receive home oxygen therapy (HOT) and noninvasive positive pressure ventilation (NPPV). Since NPPV does not involve tracheal intubation, patients are able to talk, eat, and drink as they go about their daily lives. The use of NPPV has significantly improved the life expectancy and quality of life (QOL) of patients [14]. However, patients with chronic respiratory failure receiving NPPV experience a reduced range of activity owing to respiratory symptoms and distress when wearing a mask. Patients receiving NPPV are more prone to having high carbon dioxide levels and need to learn to manage their physical conditions to a higher degree [15]. Patients with chronic respiratory failure are less likely to notice changes in acute exacerbations because they usually have respiratory symptoms. This means that they delay seeking medical attention, which may lead to a more serious condition. Hospitalization due to acute exacerbation decreases activities of daily living (ADLs) and contributes to lowering a patient's QOL [14]. Harrison et al. reported that patients with chronic respiratory failure could avoid acute exacerbations through effective self-management [16]. We hypothesized that nursing support for learning self-management in all aspects of convalescent life is necessary.

Evidence of the benefits of telenursing for patients with chronic respiratory failure receiving NPPV, who are more dependent on medical care, is limited. Therefore, this study examined the effectiveness of a telenursing intervention program that combines non-pharmacological therapies, such as symptom management and infection prevention behaviors, for patients with chronic respiratory failure receiving NPPV.

## Materials and methods

### Study design and setting

The study design was a randomized controlled trial with an intervention group using an ICT-based telenursing intervention program for patients with chronic respiratory failure receiving NPPV at home in addition to their usual care and a control group receiving usual care.

### Participants

The study participants included patients with chronic respiratory failure who received NPPV and attended respiratory outpatient clinics from September 2017 to September 2018. The inclusion criteria were as follows: chronic respiratory failure due to respiratory diseases; NPPV support; state of respiratory failure lasting more than 1 month with a partial pressure of arterial

blood carbon dioxide >45 mmHg; no specific disease and, therefore, cannot be staged; age ≥20 years; ability to use a tablet device. The exclusion criteria were as follows: inability to communicate due to cognitive impairment; inability to speak Japanese; non-attendance at respiratory outpatient clinics.

The average number of hospitalizations among NPPV practitioners in Japan was 2.0 times/year, and it was assumed that the average number of hospitalizations would improve by 0.25 times/ three months. The required sample size was calculated with a significance level of 5%, a power of 80%, and a standard deviation of the assumed average number of hospitalizations of 0.25 times/ three months. As a result, the intervention and control groups were set at 15 participants each.

## Treatment groups

The attending physician at the cooperating facility selected participants based on the eligibility criteria and explained the documents related to the study. An initial interview was conducted with participants who were interested in the study. After submitting a consent form for participation at the initial interview, the participants were enrolled in the study. Consenting participants were assigned serial numbers, consolidated, and anonymized. Research assistants assigned the participants to the intervention and control groups by fitting them to a random number table generated from the serial numbers. Baseline data, including basic attributes and primary and secondary endpoints, were measured. The intervention group was given an overview of the telenursing intervention program and asked to input their physical condition, including vital signs, symptoms, and signs, using ICT at home. At the start of the program, we visited the participants' homes to check the tablet operation and set up communication.

## Telenursing intervention program

The telenursing intervention program implemented in this study used a telenursing system (COMPAS) developed by the authors. This program was also designed as a nursing intervention program to reduce acute exacerbations by integrating self-monitoring and self-management into the lifestyles of patients with chronic respiratory failure receiving NPPV on a proactive basis according to their individual physical and mental conditions. The program consists mainly of telemonitoring and telehealth counseling using a telenursing system over three months. The tablet devices used by the participants were provided by the researcher (M. S.). Once a day, each participant answered the questions displayed on the tablet from a list of options in approximately 10 min. Data on the physical condition, including vital signs, respiratory symptoms, food intake, excretion, medication use, physical symptoms other than respiratory symptoms, and questions to the medical staff, were sent to the server and collected. The transmitted data were monitored remotely, and nursing support was determined based on the results of each question item set in advanced consultation with the physician in charge.

When a change in physical condition was observed, the condition was assessed using a videophone, telephone, or e-mail, and nursing support was provided based on the Respiratory Rehabilitation Manual [17]. The participants could input data into the tablet terminal from 08:00 to 12:00, and the nursing interventions were conducted as needed from 13:00 to 17:00 for approximately 30 min.

In cases of failure to enter data on the tablet device for more than 3 days, the participant was contacted by phone to determine the cause. If the problem could not be solved, we immediately called back and addressed the issue as soon as possible.

Since the tablet device was provided by the researchers, they were collected following the completion of the intervention program, and all communication costs for this study were borne by the researchers. This program was supervised by several professionals during its

development, including respiratory medicine specialists, nurses engaged in respiratory medicine, physical therapists, and occupational therapists.

## Telenursing system (COMPAS)

This system was developed based on the researcher's preliminary investigation [15] and literature review, and the Web-based program comprised a patient site and a researcher site. The tablet of the patient site contains "daily physical status record," "video phone (Skype)," and "respiratory rehabilitation information (PDF, YouTube videos)." The researcher's site included a list of the participants' telemonitoring data and the ability to enter comments. The researcher could review the daily records of the participants and record the changes over time on a summary sheet (S1 and S2 Appendices).

The server was managed by a system production and development company, which encrypted the transmitted data with Secure Socket's Layer encryption. The researcher and the physician in charge of the cooperating medical institution were the only ones authorized to access the server. Health information on the server in this study was not integrated into the hospital's electronic health record, due to hospital security controls. The communication technology used was Japanese cell phone communication. Near Field Communication (NFC) function was used to collect vital signs data. NFC is a short-range wireless communication technology that uses a frequency of 13.56 MHz and has a communication distance of approximately 10 cm. The NFC function assisted the participants with inputting data on vital signs. The NFC function facilitates the automatic transfer of patient vital sign data from the measurement device to the tablet. Additionally, the system does not include an alert function for clinicians. Therefore, the system was designed to allow constant communication between clinicians and researchers.

## Survey method and contents

**Characteristics of the participants.** The following characteristics of the participants were collected at the time of registration: age, sex, primary diagnosis, family composition, ADL, level of care required, implementation status of NPPV, implementation status of HOT, smoking history, and availability of social resources. The pulmonary ADL (P-ADL) was used as the ADL assessment scale. P-ADL was developed in Japan as a respiratory disease-specific ADL scale because the standard ADL scale does not accurately capture ADL impairment due to breathlessness in respiratory diseases [18].

**Study endpoints.** The primary endpoint of the study was the number of acute exacerbations in the past 3 months. For the number of acute exacerbations, the number of outpatient visits, hospital admissions, and hospital days was collected. The secondary endpoints included the St. George's Respiratory Questionnaire (SGRQ) score [19], Self-Care Agency Questionnaire (SCAQ) score [20], Euro QOL 5 Dimension (EQ-5D) score [21], pulmonary function tests, and 6-min walk distance (6MWD). These items were evaluated at enrollment and follow-up (3 months after enrollment).

The SGRQ was developed as a COPD-specific health-related quality-of-life assessment instrument, but has since been tested for reliability and validity for a variety of respiratory diseases. The scale is designed to assess changes over time in respiratory symptoms, such as cough, dyspnea, and wheezing, as well as social and psychological effects of these symptoms. It has excellent ability to depict changes due to medical intervention. It is a self-administered questionnaire with 50 questions, and the score is calculated by summing the components of Symptom, Activity, and Impact. "Symptom" indicates the frequency and severity of symptoms such as cough, phlegm, wheezing, and dyspnea. "Impact" represents the social activities and psychological disturbances affected by COPD. Each score ranges from 0 to 100, with "0"

indicating no impairment and higher values indicating greater impairment. A cutoff value of 4 points indicates the minimum, clinically significant change.

The SCAQ scale assesses the self-care abilities of chronically ill patients. It consists of four subscales with a total of 29 items: 10 items for "acquisition and continuation of health care methods," 7 items for "adjustment of physical condition," 7 items for "interest in health care," and 5 items for "acquisition of effective support". Each questionnaire item is on a 5-point Likert scale: 5 for "yes," 4 for "somewhat yes," 3 for "neither yes nor no," 2 for "somewhat no," and 1 for "no." It is a self-administered questionnaire, and scores can be calculated for the entire questionnaire and for each construct. The maximum score is 145 points, with higher scores indicating greater self-care ability. No reference score is defined.

The EQ-5D is a comprehensive quality of life scale, developed in 1987 by EuroQol, a research group consisting of educational and medical facilities in the United Kingdom and Finland, and has been measured for various diseases. The five items include degree of mobility, personal care, daily activities, pain/discomfort, and anxiety/concern. It is a self-administered questionnaire and the results are used to calculate a standardized health score ranging from "perfect health = 1" to "death = 0".

## Statistical analysis

The basic attributes of the participants were subjected to simple tabulation followed by statistical analysis. The differences between the endpoints at baseline and 3 months after enrollment were then calculated and used to compare the intervention and control groups. The Mann–Whitney U test was used for comparison. The number and content of interventions were tabulated. We used a personal computer with Microsoft Windows 8.1 operating system (Microsoft Corp., Redmond, WA, USA), and statistical analysis was performed using IBM SPSS Statistics ver. 21.0 software (IBM Corp., Armonk, NY, USA). The significance level was set at two-tailed $p < .05$. The intention-to-treat (ITT) analysis method was employed in this study.

## Ethical considerations

The study protocol was approved by the Ethics Committee of the Tohoku University Graduate School of Medicine (approval number: 2017-1-400) and adhered to the ethical principles of the Declaration of Helsinki. This study was conducted in compliance with the CONSORT guidelines. The trial registration number is UMIN-CTR (UMIN000027657). The study details, including an overview of the study objectives and other information, how to withdraw cooperation, protection of personal information, how to operate the tablet, encryption of transmitted data, and restrictions on access to the server, were adequately explained to the participants, and consent to participate was obtained orally and in writing.

The study provisions included the tablet and communication costs associated with the study but not the electricity bill for using the tablet. Consolidated anonymization was used to protect participants' personal information. The intervention program was conducted in a private room where privacy was maintained, and non-researcher access to the room was restricted during data viewing, phone calls, and video calls. The researcher always logged out of the telenursing system when leaving the computer terminal.

## Results

### Participant inclusion

In total, 33 patients met the participation criteria at the recruiting facility, all of whom were referred by the facility. Of the 33 participants referred by the facilities, 31 (n = 15, intervention

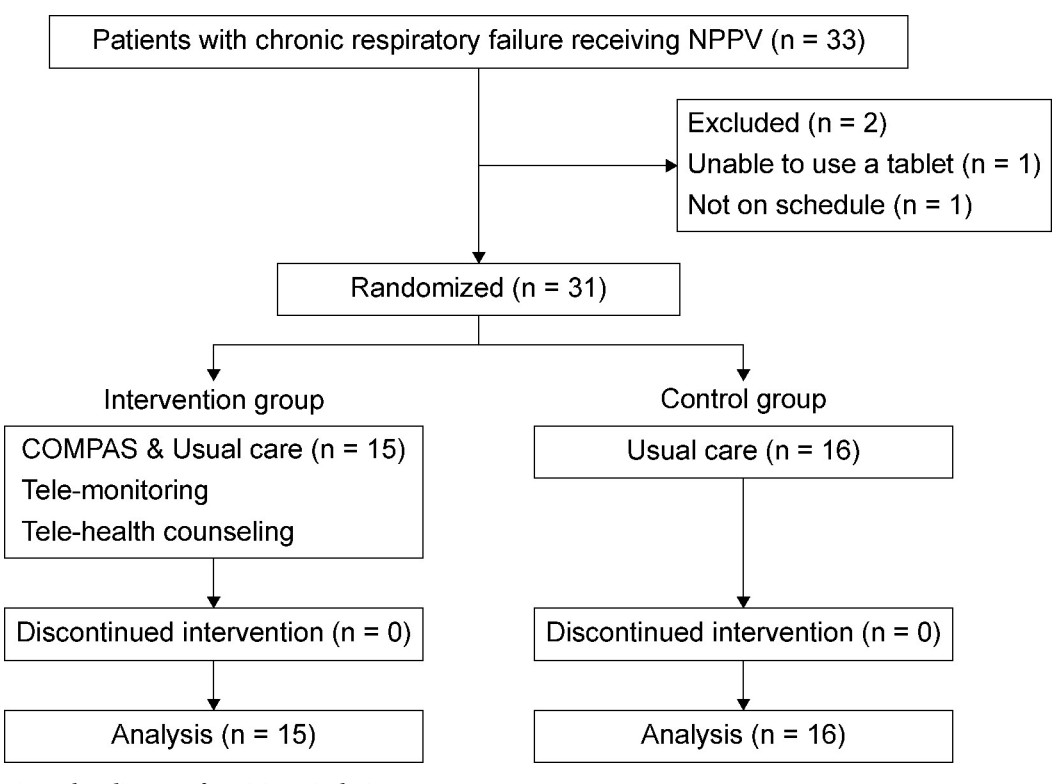

**Fig 1. Flow diagram of participant inclusion.**

group; n = 16, control group) agreed to participate in the study. The two participants who did not agree to participate cited the following reasons: "too old to use a tablet device" and "unable to participate because of the survey schedule." No participants dropped out after registration; thus, 31 participants were included in the analysis. The flowchart of participant inclusion is shown in Fig 1.

## Characteristics of the study participants

The characteristics of the participants are shown in Table 1. There were nine men (60.0%) and six women (40.0%) in the intervention group and nine men (56.3%), and seven women (43.8%) in the control group. The mean age of all participants was 73.0 (standard deviation [SD]: 10.2) years; mean height, 155.8 (SD: 9.5) cm; mean weight, 60.5 (SD: 12.7) kg; and mean body mass index, 25.1 (SD: 5.6) kg/m$^2$.

 COPD was the main cause of chronic respiratory failure in 13 patients (41.9%), followed by alveolar hypoventilation syndrome in 10 (32.3%), sequelae of pulmonary tuberculosis in six (19.4%), and scoliosis and spinal caries in one each. All patients were prescribed NPPV only at night during bedtime. Approximately half (n = 16, 51.6%) of the patients were on HOT. There were 19 past smokers (61.3%) and 30 current non-smokers (96.8%). Exactly 28 patients (90.3%) were living with family members, and 13 (41.9%) were using social resources. The mean use duration of NPPV and HOT was 3.0 (SD: 3.7) years and 2.0 (SD: 1.6) years, respectively. The P-ADL score was 169.5 (SD: 22.9) points. There were no significant differences in the characteristics of the participants between the intervention and control groups at enrollment.

**Table 1. Characteristics of the study participants (N = 31).**

| Evaluation items | Total | | Intervention group | | Control group | |
|---|---|---|---|---|---|---|
| | N = 31 | | n = 15 | | n = 16 | |
| Sex, n (%) | | | | | | |
| Male | 18 | (58.1) | 9 | (60.0) | 9 | (56.3) |
| Female | 13 | (41.9) | 6 | (40.0) | 7 | (43.8) |
| Age (years), mean±SD | 73.0 | ±10.2 | 72.5 | ±10.9 | 73.6 | ±10.0 |
| Body data, mean±SD | | | | | | |
| Length (cm) | 155.8 | ±9.5 | 153.8 | ±0.1 | 157.4 | ±10.7 |
| Weight (kg) | 60.5 | ±12.7 | 60.9 | ±15.3 | 60.2 | ±10.6 |
| BMI | 25.0 | ±5.6 | 25.9 | ±7.1 | 24.4 | ±4.2 |
| Duration of oxygen therapy, mean±SD | | | | | | |
| NPPV (years) | 3.0 | ±3.7 | 2.9 | ±3.3 | 3.2 | ±4.1 |
| HOT (years) | 2.0 | ±1.6 | 1.7 | ±1.6 | 2.3 | ±1.6 |
| ADL, mean±SD | | | | | | |
| P-ADL (points) | 169.5 | ±22.92 | 167.1 | ±23.0 | 171.8 | ±23.3 |
| Primary disease, n (%) | | | | | | |
| COPD | 13 | (41.9) | 6 | (40.0) | 7 | (43.8) |
| Pulmonary cell hypoventilation syndrome | 10 | (32.3) | 5 | (33.3) | 5 | (31.3) |
| Pulmonary tuberculosis sequelae | 6 | (19.4) | 2 | (13.3) | 4 | (25.0) |
| Scoliosis | 1 | (3.2) | 1 | (6.7) | | |
| Spinal caries | 1 | (3.2) | 1 | (6.7) | | |
| NPPV treatment schedule, n (%) | | | | | | |
| Nocturnal | 31 | (100.0) | 15 | (100.0) | 16 | (100.0) |
| HOT, n (%) | 16 | (51.6) | 8 | (53.3) | 8 | (50.0) |
| Current smoking, n (%) | 1 | (3.2) | 0 | (0.0) | 1 | (6.3) |
| Past smoking, n (%) | 19 | (61.3) | 10 | (66.7) | 9 | (56.3) |
| With family, n (%) | 28 | (90.3) | 13 | (86.7) | 15 | (93.8) |
| Social resources, n (%) | 13 | (41.9) | 4 | (26.7) | 9 | (56.3) |

BMI, body mass index; ADL, activities of daily living; SD, standard deviation; HOT, home oxygen therapy; NPPV, noninvasive positive pressure ventilation; COPD, chronic obstructive pulmonary disease

## Study endpoints at enrollment

The primary and secondary endpoints at enrollment were compared between the intervention and control groups and presented in Table 2 (S3 Appendix) using the Mann-Whitney U test.

There was no significant difference in the number of outpatient visits, hospitalizations, or days of hospitalization between the intervention and control groups at enrollment. Overall, there was no significant difference in the secondary endpoints between the intervention and control groups at enrollment.

## Differences between the endpoints at baseline and 3 months post-enrollment

Differences between the endpoints at baseline and 3 months post-enrollment were compared (Table 3, Fig 2, S4 and S5 Appendices) by Mann-Whitney U test. The intervention group had significantly fewer hospitalizations (r = .38; p = .037) and shorter hospital stays (r = .39; p = .031) than the control group. There was a significant increase in the number of unscheduled outpatient visits in the intervention group compared to the control group (r = 36; p = .045).

**Table 2. Assessment scores at enrollment.**

| Measure | Intervention group (n = 15) | | Control group (n = 16) | | ES *(r)* | p |
|---|---|---|---|---|---|---|
| | Median | IQR | Median | IQR | | |
| Number of hospitalizations in the past 3 months (times) | 0.0 | 0.0–0.0 | 0.0 | 0.0–0.0 | .20 | .262 |
| Number of days hospitalized in the past 3 months (days) | 0.0 | 0.0–0.0 | 0.0 | 0.0–0.0 | .22 | .222 |
| Number of unscheduled outpatient visits in the past 3 months (times) | 0.0 | 0.0–0.0 | 0.0 | 0.0–0.75 | .25 | .164 |
| SGRQ | | | | | | |
| Symptom | 44.2 | 18.9–61.6 | 32.7 | 24.9–59.0 | .07 | .693 |
| Activity | 67.1 | 60.4–92.5 | 66.6 | 59.5–80.4 | .17 | .341 |
| Impact | 29.1 | 14.2–53.4 | 29.8 | 21.9–39.4 | .02 | .906 |
| Total Score★ | 44.3 | 31.0–63.7 | 46.4 | 34.5–51.3 | .04 | .813 |
| EQ–5D | 0.8 | 0.5–0.8 | 0.8 | 0.6–0.8 | .09 | .607 |
| SCAQ | | | | | | |
| Acquisition and continuation of healthcare methods | 43.0 | 37.0–47.0 | 39.0 | 34.5–47.8 | .12 | .513 |
| Adjustment of physical condition | 30.0 | 29.0–33.0 | 31.0 | 28.3–33.8 | .07 | .706 |
| Interest in healthcare | 33.0 | 29.0–35.0 | 33.5 | 31.3–33.5 | .07 | .698 |
| Obtaining effective support | 22.0 | 17.0–24.0 | 21.5 | 19.3–24.8 | .12 | .498 |
| Total score★ | 127.0 | 117.0–136.0 | 124.5 | 119.0–137.0 | .02 | .905 |
| Pulmonary function | | | | | | |
| %Lung capacity (%) | 62.7 | 45.0–77.9 | 58.7 | 46.4–72.0 | .03 | .935 |
| 1 s rate (%) | 73.1 | 50.0–78.1 | 60.5 | 50.4–74.8 | .17 | .584 |
| Exercise tolerance | | | | | | |
| 6-min walk distance (m) | 320.0 | 173.0–370.0 | 290.0 | 135.0–361.8 | .11 | .749 |

Mann–Whitney *U* test ★SGRQ Total Score and SCAQ Total Score are medians of the sum rather than the sum of partial scores.

IQR, interquartile range; ES, effect size; SGRQ, St. George's Respiratory Questionnaire; SCAQ, Self-Care Agency Questionnaire

Group comparisons of the differences in secondary endpoints revealed significantly lower SGRQ "activity" scores in the intervention group than in the control group (r = .37; p = .039). This indicates that the intervention group had a higher QOL with regard to activity. The SCAQ total scores of the intervention group were significantly higher than those in the control group (r = .38; p = .034), indicating a higher capacity for self-care.

## Telehealth counseling

The total number of telehealth counseling sessions throughout the study period was 35, with an average of 2.3 consultations per participant (Table 4). There were 13 videophone consultations, 12 telephone consultations, and 10 e-mail consultations. There were 16 telehealth counseling sessions via remote monitoring. The most common reason for consultation was the deterioration of vital signs (n = 6). The number of interventions based on the patient's request was 19. There were six consultations regarding physical activity and four regarding dyspnea.

## Discussion

This study examined the effectiveness of a telenursing intervention program in reducing exacerbations in patients with chronic respiratory failure receiving NPPV using a randomized controlled trial. The telenursing intervention program used in this study consisted of telemonitoring and telehealth counseling.

**Table 3. Differences between the endpoints at baseline and 3 months post-enrollment.**

| Measure | Intervention group (n = 15) | | Control group (n = 16) | | ES *(r)* | p |
|---|---|---|---|---|---|---|
| | **Median** | **IQR** | **Median** | **IQR** | | |
| Number of hospitalizations in the past 3 months (times) | 0.0 | 0.0–0.0 | 0.0 | 0.0–0.8 | .38 | .037* |
| Number of days hospitalized in the past 3 months (days) | 0.0 | 0.0–0.0 | 0.0 | 0.0–7.5 | .39 | .031* |
| Number of unscheduled outpatient visits in the past 3 months (times) | 0.0 | 0.0–1.0 | 0.0 | 0.0–0.0 | .36 | .045* |
| SGRQ | | | | | | |
| Symptom | -3.9 | -12.2–8.5 | -7.3 | -15.0–6.7 | .15 | .395 |
| Activity | -5.9 | -18.0–0.0 | 0.0 | -5.0–6.9 | .37 | .039* |
| Impact | 0.0 | -10.5–10.8 | -0.5 | -7.3–2.5 | .13 | .477 |
| Total Score★ | -0.5 | -12.6–4.9 | -1.5 | -4.8–1.1 | .04 | .843 |
| EQ–5D | 0.0 | -0.2–0.1 | 0.0 | -0.1–0.0 | .06 | .722 |
| SCAQ | | | | | | |
| Acquisition and continuation of healthcare methods | -1.0 | -2.0–4.0 | -3.0 | -6.8–0.5 | .29 | .113 |
| Adjustment of physical condition | 1.0 | -1.0–2.0 | -1.0 | -2.0–2.5 | .22 | .225 |
| Interest in healthcare | 0.0 | -1.0–3.0 | 0.0 | -2.5–0.8 | .22 | .231 |
| Obtaining effective support | 1.0 | -1.0–6.0 | -1.0 | -2.0–0.8 | .33 | .067 |
| Total score★ | 1.0 | -1.0–8.0 | -5.0 | -9.0–3.0 | .38 | .034* |
| Pulmonary function | | | | | | |
| %Lung capacity (%) | 1.8 | -3.2–8.9 | 1.1 | -1.8–4.2 | .00 | 1.000 |
| One second rate (%) | -1.9 | -4.6–1.1 | 0.0 | -3.0–8.0 | .22 | .226 |
| Exercise tolerance | | | | | | |
| 6-min walk distance (m) | -50.4 | -212.5–686.6 | -123.5 | -720.6–21.4 | .35 | .050 |

Mann-Whitney *U* test

*p < .05

★SGRQ Total Score and SCAQ Total Score are medians of the sum rather than the sum of partial scores.

ES, effect size; SGRQ, St. George's Respiratory Questionnaire; SCAQ, Self-Care Agency Questionnaire

The number of hospitalizations for acute exacerbations and the duration of hospital stay at the 3-month follow-up were lower in the intervention group, suggesting that the proposed telenursing intervention program is effective in reducing acute exacerbations in patients with chronic respiratory failure receiving NPPV. Previous studies using ICT have similarly reported a reduction in the number of hospitalizations due to acute exacerbations in patients with chronic respiratory failure [6, 11]. As there are no reports limited to patients with chronic respiratory failure receiving NPPV, the results of this study could be an important finding.

### Effectiveness of telemonitoring

We found that patients with chronic respiratory failure receiving NPPV routinely experience respiratory symptoms, symptoms of hypercapnea, and pressure of the face mask during their recuperation. Some participants recognized the various symptoms as age-related changes or those of the common cold and took necessary coping actions, such as resting and monitoring the symptoms. Some participants thought that they should not attempt outpatient visits outside of their regular appointments and chose to wait until the next regular outpatient visit, despite having symptoms of an acute exacerbation. These results indicate that although signs of an acute exacerbation were detected, participants did not take prompt medical attention. The SCAQ score, which indicates self-care ability, was significantly higher in the intervention group than in the control group, suggesting that the intervention program may have improved the self-management ability of the participants.

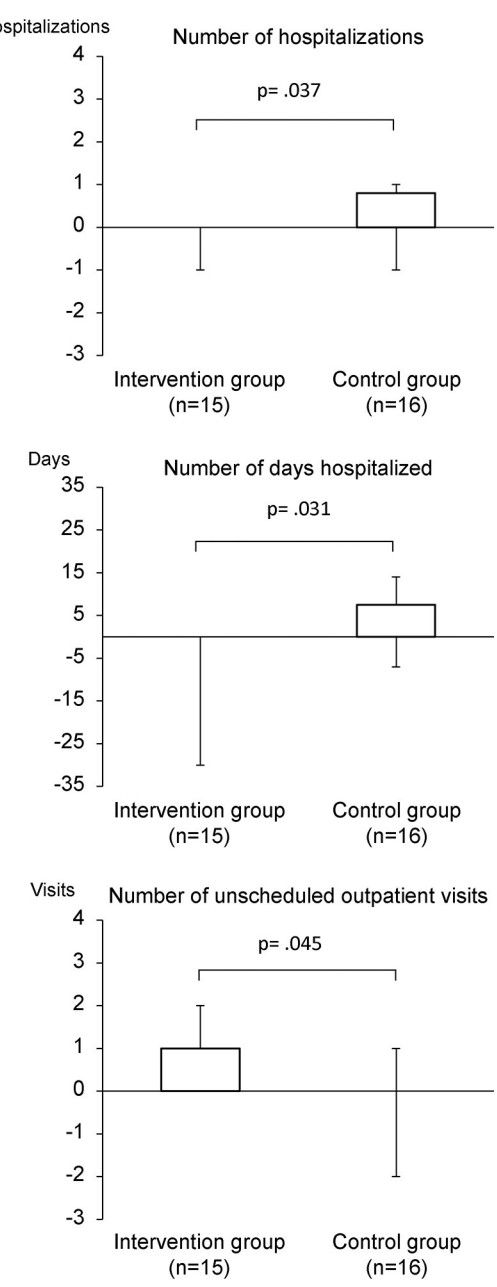

**Fig 2. Differences between the primary endpoints at baseline and 3 months post-enrollment.**

When the participants experienced a change in physical condition, they were instructed to contact the researcher for a telehealth counseling session. The researcher confirmed the results of the telemonitoring and explained the meaning of the symptoms and coping behaviors, which may have led the participant to learn the meaning of the change in the physical condition and visit an outpatient clinic. According to previous studies using ICT, the number of visits to the emergency room was reduced because patients were able to receive direct instructions on medical care to be provided at home from medical personnel as a result of telemonitoring [22–24]. In a previous study, a participant had type II chronic respiratory failure, which is a complex condition that requires immediate and accurate medical intervention [14];

**Table 4. Data on telehealth counseling.**

| | |
|---|---|
| Number of health consultations and information provided via videophone, etc. | |
| Total | 35 |
| Average number of cases per person | 2.3 |
| Details | |
| 1) Videophone | 13 |
| 2) Phone | 12 |
| 3) E-mail | 10 |
| Contents of health consultation and information provided by videophone, etc. | |
| 1. Telemonitoring intervention | |
| Total | 16 |
| Details | |
| 1) Deterioration of vital signs | 6 |
| 2) Deterioration of non-respiratory symptoms | 4 |
| 3) Deterioration of respiratory symptoms | 1 |
| 4) Delayed input | 5 |
| 2. Intervention at the patient's request | |
| Total | 19 |
| Details | |
| 1) Physical activity | 6 |
| 2) Dyspnea | 4 |
| 3) NPPV device or mask | 3 |
| 4) Eating | 2 |
| 5) Other | 4 |

thus, the attending physician instructed the participant to seek immediate medical attention if any change in the physical condition indicating an acute exacerbation was observed. Signs of an acute exacerbation include not only the appearance of clear respiratory symptoms but also sensations that are somehow different from the norm [14], wherein "sensations that are somehow different from the norm" can refer to malaise or issues with body movement that occur during a physical illness.

It is difficult to link these subtle sensations to therapeutic actions, and many participants decided to wait and observe symptom progression without consulting a doctor. Symptoms of acute exacerbations, such as shortness of breath, sputum volume, and changes in sputum coloration, rarely appear more than once. As they are short-lived, they are not easily judged to be acute exacerbations [14]. However, early detection and treatment of signs of acute exacerbations are important to prevent progression to acute exacerbations that require hospitalization. A previous study showed that the number of nurse visits in the telemonitoring group increased, although the hospitalization rate of patients decreased [25]. Telemonitoring is considered to be important not only for the early detection of acute exacerbations but also for connecting patients to medical interventions at an early stage and reducing exacerbations.

Patients with type II respiratory failure are treated with pharmacotherapy and non-pharmacological therapies, such as respiratory rehabilitation from the stable stage. The emergence of acute exacerbations is related to complex pathological conditions, such as $CO_2$ narcosis and pulmonary heart disease. Moreover, a delay in the detection of the signs of acute exacerbations and failure to respond to them can directly lead to deterioration of the prognosis [14]. Nurses who provide telenursing to patients with type II respiratory failure should monitor the results of telemonitoring, collaborate with the attending physician, and carefully consider the method of medical intervention.

## Effectiveness of telehealth counseling

Generally, patients undergoing NPPV receive self-management support from medical staff on how to manage NPPV equipment and knowledge and skills for medical treatment. Then, they are discharged home. However, in recent years, there have been many older patients with cognitive decline in the latter half of their lives, and there are cases where it is difficult for them to utilize the knowledge and skills they have received at home. In this program, the participants entered information about NPPV management, medication, diet, excretion, and physical activity, and the researcher provided telehealth counseling for questions and concerns regarding daily life. As a result, the participants were able to reconfirm the points to keep in mind during their medical treatment and may have been able to maintain their health management behavior.

Patients with respiratory diseases who continue appropriate self-management have reduced shortness of breath, improved health-related QOL, and decreased number of respiratory-related hospitalizations [26]. Since telenursing is conducted face-to-face via videoconference, it is easier for patients to feel at ease and build a trusting relationship [27]. Telehealth counseling provided the participants with a strong sense of connection with the researcher. In addition, as it was easy to understand the living conditions of the participants, it was possible to assess their self-care abilities and provide tailored information, although they were living far away.

Patients living with NPPV at home face challenges and questions they do not face during hospitalization. In this study, telehealth counseling allowed participants to respond immediately to any changes in symptoms or questions about treatment. As a result, they could easily cope with the situation and acquire new knowledge and skills from the experience. For effective self-management support, behavioral change theory is used, and it is important to provide education and explanation promptly when the patient is conscious [28]. In this program, the timing of the telehealth counseling was not decided in advance, and the program might have promoted smoother behavioral changes in patients by providing counseling when they needed it.

Self-management education using ICT has been practiced in Japan and abroad and has been reported to be effective in reducing the number of hospitalizations and maintaining adherence [29, 30]. In this study, the intervention group had higher SCAQ scores and showed better maintenance of self-care capability than the control group. Especially, it is inferred that new knowledge and skills were added by self-management support at the appropriate time. In addition, acute exacerbation may have been reduced by the continuation of appropriate health management strategies, such as in terms of food, excretion, and medication, during treatment.

## Future development of telenursing

The causes of acute exacerbations in patients with chronic respiratory failure mainly include respiratory infections and environmental factors; however, in approximately 30% of cases, the cause remains unknown [14]. Patient-reported exacerbations were the most frequently reported, and patients need to learn how to assess and cope with acute exacerbations by themselves [14]. We demonstrated that a telenursing intervention program consisting of telemonitoring and telehealth counseling sessions reduced the number of hospitalizations and duration of hospital stay. Thus, we believe that telenursing is an effective way to help patients recognize the signs of acute exacerbations and provide coping strategies at an early stage.

To effectively prevent the severity of acute exacerbations in patients with chronic respiratory failure, it is necessary to improve the quality of regular medical care and outpatient nursing care and to promote telenursing. In recent years, setting up an action plan according to the

patient's symptoms has been considered effective in preventing acute exacerbations of COPD [31, 32]. In particular, when patients become aware of signs of an acute exacerbation, they should take antimicrobial agents or oral steroids as instructed by their physicians [31]. More than half of the participants in this study did not have an action plan, and the physician instructed the patients to attend an outpatient visit promptly at the first sign of an acute exacerbation. However, there were times when the participants avoided visiting the outpatient clinic because of the burden of travel expenses, time, and physical exhaustion. Additionally, due to security issues within the hospital, the patient health information obtained in this study was not integrated into the hospital's electronic medical record. We believe that this is one reason that, although various tele-nursing systems are currently being developed, their implementation has been hindered.

Thus, it is necessary to integrate existing medical care and telenursing systems to complement each other so that the burden on patients can be reduced and the severity of acute exacerbations can be prevented.

## Limitations of the study

This study had some limitations. First, as the participants were mainly older adults aged ≥60 years, it took some time to learn how to use the tablet devices. In addition, although the telenursing system used in this program was created for older adults, several home visits were required because the participants could not address issues regarding the communication environment or the applications by themselves. This may have had some impact on the evaluation of this study. Future studies should improve the communication environment and the methods of the system to make it more convenient for older patients. In comparison with a previous study [28], the follow-up period in this study (3 months) may have been too short for the participants to make behavioral changes in their lifestyles. Thus, the longer-term effects of this intervention should be further investigated. Though employing the ITT analysis, the missing values were not evaluated since there were no missing values in the endpoints. In addition, this study did not adjust for covariates in the multivariate analysis, which may have resulted in biased results.

## Conclusion

The effectiveness of a telemedicine intervention program to reduce exacerbations in patients with chronic respiratory failure receiving NPPV was evaluated in this randomized controlled trial. After 3 months of exposure to the telenursing intervention program, both the number of hospitalizations for acute exacerbations and the number of hospital days decreased. In addition, there was an improvement in the self-care ability of the patients in the intervention group. Therefore, the telemedicine intervention program may reduce exacerbations in patients with chronic respiratory failure receiving NPPV.

## Supporting information

**S1 Checklist. CONSORT 2010 checklist of information to include when reporting a randomised trial***.
(DOC)

**S1 Appendix. Overview of the telenursing system (COMPAS).**
(TIF)

**S2 Appendix. Screen display of the telenursing system (COMPAS).**
(TIF)

**S3 Appendix. Primary endpoints at enrollment (Baseline).**
(DOCX)

**S4 Appendix. Primary endpoints after 3 months of enrollment.**
(DOCX)

**S5 Appendix. Differences between the baseline and 3 months post-enrollment for the primary endpoints.**
(DOCX)

**S1 File.**
(DOCX)

**S2 File.**
(DOC)

## Acknowledgments

We would like to express our sincere gratitude to all the participants and their families who cooperated in this survey. We would also like to express our gratitude to the doctors, nurses, and other co-medical staff at the facilities that cooperated in the survey.

## Author Contributions

**Conceptualization:** Makoto Shimoyama, Shiori Yoshida, Chikako Takahashi, Mizue Inoue, Naoko Sato, Fumiko Sato.

**Data curation:** Makoto Shimoyama, Fumiko Sato.

**Formal analysis:** Makoto Shimoyama, Fumiko Sato.

**Funding acquisition:** Makoto Shimoyama.

**Investigation:** Makoto Shimoyama, Fumiko Sato.

**Methodology:** Makoto Shimoyama, Shiori Yoshida, Chikako Takahashi, Mizue Inoue, Naoko Sato, Fumiko Sato.

**Project administration:** Makoto Shimoyama.

**Resources:** Makoto Shimoyama.

**Software:** Makoto Shimoyama.

**Supervision:** Shiori Yoshida, Chikako Takahashi, Mizue Inoue, Naoko Sato, Fumiko Sato.

**Validation:** Makoto Shimoyama.

**Visualization:** Makoto Shimoyama.

**Writing – original draft:** Makoto Shimoyama, Shiori Yoshida, Chikako Takahashi, Mizue Inoue, Naoko Sato, Fumiko Sato.

**Writing – review & editing:** Makoto Shimoyama.

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
