## [Decision Letter · Decision Letter 0]

6 Sep 2022

PONE-D-22-13518Effectiveness of a telenursing intervention program in reducing exacerbations in patients with chronic respiratory failure receiving noninvasive positive pressure ventilation: A randomized controlled trialPLOS ONE

Dear Dr. Shimoyama,

Thank you for submitting your manuscript to PLOS ONE. After careful consideration, we feel that it has merit but does not fully meet PLOS ONE’s publication criteria as it currently stands. Therefore, we invite you to submit a revised version of the manuscript that addresses the points raised during the review process.

 Please note that we have only been able to secure a single reviewer to assess your manuscript. We are issuing a decision on your manuscript at this point to prevent further delays in the evaluation of your manuscript. Please be aware that the editor who handles your revised manuscript might find it necessary to invite additional reviewers to assess this work once the revised manuscript is submitted. However, we will aim to proceed on the basis of this single review if possible. The reviewer has suggested opportunities to significantly improve the rigour of your statistical analyses and to clarify aspects of the data. Please respond carefully to all the points they have raised when preparing your revisions.

We look forward to receiving your revised manuscript.

Kind regards,

Jamie Males

Editorial Office

PLOS ONE

Journal Requirements:

2. Thank you for submitting your clinical trial to PLOS ONE and for providing the name of the registry and the registration number. The information in the registry entry suggests that your trial was registered after patient recruitment began. PLOS ONE strongly encourages authors to register all trials before recruiting the first participant in a study.

1) your reasons for your delay in registering this study (after enrolment of participants started);

2) confirmation that all related trials are registered by stating: “The authors confirm that all ongoing and related trials for this drug/intervention are registered”.

Reviewers' comments:

Reviewer's Responses to Questions

**Comments to the Author**

1. Is the manuscript technically sound, and do the data support the conclusions?

Reviewer #1: No

2. Has the statistical analysis been performed appropriately and rigorously? 

Reviewer #1: No

3. Have the authors made all data underlying the findings in their manuscript fully available?

Reviewer #1: No

4. Is the manuscript presented in an intelligible fashion and written in standard English?

Reviewer #1: Yes

5. Review Comments to the Author

Reviewer #1: PONE-D-22-13518: statistical review

SUMMARY. This is a study on the effectiveness of a telenursing intervention program in patients receiving noninvasive positive pressure ventilation (NPPV) using a randomized controlled trial. The core statistical analysis relies on a battery of Mann-Whitney U tests that compare the treatment and the control groups in two different points in time. I have a couple of serious concerns about this paper (see major issues 1 and 2 below). I also append some specific points that should be addressed.

MAJOR ISSUES

1. The outcomes of the treatment and the control group are separately compared by two independent batteries of Mann-Whitney tests. This approach ignores the longitudinal structure of the data: individuals are observed at two different occasions. I would suggest performing a single battery of tests on the temporal differences of the outcomes. Specifically, the authors should first compute the individual difference between the value of each outcome at the end of the follow-up and the value at baseline. They should then perform a test that compares the differences in the treatment group with the differences in the control group.

2. Table 2 seems to indicate that the primary outcomes (Number of hospitalizations in the past 3 months, Number of days hospitalized in the past 3 months, Number of unscheduled outpatient visits in the past 3 months) do not vary much within each sample. How is this possible? These data need some clarification. In addition, could the author please provide the single numbers? See also specific point 4.

SPECIFIC POINTS.

1. Tables 2 and 3: the total scores of SGRQ and SCAQ are not the sums of the partial scores. Please clarify.

2. Lines 252-268 are just a description of Table 2 and could be shortened,

3. Lines 120-122: "Research assistants ... numbers". This sentence is unnecessarily repeated on line 127.

4. Data are not provided. Data should be provided as a supplementary file or made available in a public repository.

6. PLOS authors have the option to publish the peer review history of their article (what does this mean?). If published, this will include your full peer review and any attached files.

Reviewer #1: No

---

## [Author Response · Author response to Decision Letter 0]

7 Nov 2022

November 7, 2022

Jamie Males

Executive Editor

PLOS ONE

Dear Jamie Males:

Thank you for inviting us to submit a revised draft of our manuscript entitled, “Effectiveness of a telenursing intervention program in reducing exacerbations in patients with chronic respiratory failure receiving noninvasive positive pressure ventilation: A randomized controlled trial” to PLOS ONE. We also appreciate the time and effort you and each of the reviewers have dedicated to providing insightful feedback on ways to strengthen our paper. Thus, it is with great pleasure that we resubmit our article for further consideration. We have incorporated changes that reflect the detailed suggestions you have graciously provided. We also hope that our edits and the responses we provide below satisfactorily address all the issues and concerns you and the reviewers have noted.

We hope that the revised manuscript is now suitable for publication in your journal.

Thank you for your consideration. We look forward to hearing from you.

Sincerely,

Makoto Shimoyama

School of Nursing

Miyagi University

1-1 Gakuen, Taiwa-cho, Kurokawa-gun, Miyagi, 981-3298, Japan

Telephone: +81 22 377 8277

To facilitate your review of our revisions, the following are point-by-point responses to the questions and comments delivered in your letter dated September 7, 2022

JOURNAL REQUIREMENTS：

RESPONSE: Thank you for your suggestion. We have reviewed and confirmed the style requirements again. 

2. Thank you for submitting your clinical trial to PLOS ONE and for providing the name of the registry and the registration number. The information in the registry entry suggests that your trial was registered after patient recruitment began. PLOS ONE strongly encourages authors to register all trials before recruiting the first participant in a study.

RESPONSE: Thank you for your suggestion. We registered this study as a clinical trial with UMIN-CTR. We checked UMIN-CTR again and found that the clinical trial was registered on June 6, 2017. Recruitment of participants for the study began in September 2017, so registration was not delayed. For these reasons, no additional information has been added.

Please check the following URL;

https://center6.umin.ac.jp/cgi-open-bin/ctr/ctr.cgi?function=brows&action=brows&recptno=R000031552&type=summary&language=J　

Please check the "Management information" in the web page provided. The "Registered date" was June 06, 2017. We initially thought that you wanted confirmation on “Date of disclosure of the study information,” which was June 20, 2019; however, this was not the registration date.

Reviewer #1: PONE-D-22-13518: statistical review

Reviewers' comments:

MAJOR ISSUES

1. The outcomes of the treatment and the control group are separately compared by two independent batteries of Mann-Whitney tests. This approach ignores the longitudinal structure of the data: individuals are observed at two different occasions. I would suggest performing a single battery of tests on the temporal differences of the outcomes. Specifically, the authors should first compute the individual difference between the value of each outcome at the end of the follow-up and the value at baseline. They should then perform a test that compares the differences in the treatment group with the differences in the control group.

RESPONSE: Thank you for your suggestion. The details of the analysis were omitted to simplify the presentation of the results. The data has been re-analyzed, and Table 3 has been modified to show the comparison test of the difference between the intervention and target groups. 

The research methodology and results have been revised as follows: 

1. Research Methods 

・Page 11, lines 187–190: "For baseline values, the endpoints in the intervention and control groups were compared at the time of enrollment. The differences between the endpoints at baseline and 3 months after enrollment were then calculated, and the differences were used to compare the intervention and control groups. Mann-Whitney U test was used for comparison." 

2. Results

・Table 3 has been modified to compare the differences between values at baseline and at 3 months after enrollment. 

・The description of Table 3 within the results has been modified. 

2. Table 2 seems to indicate that the primary outcomes (Number of hospitalizations in the past 3 months, Number of days hospitalized in the past 3 months, Number of unscheduled outpatient visits in the past 3 months) do not vary much within each sample. How is this possible? These data need some clarification. In addition, could the author please provide the single numbers? See also specific point 4.

RESPONSE: Thank you for your suggestion. Per your observation, the primary outcome data were not clear. We have summarized the values of the primary outcome for the participants in each group in tables and provided them as Supplementary Files 1-3. 

Specific points

1. Tables 2 and 3: the total scores of SGRQ and SCAQ are not the sums of the partial scores. Please clarify.

RESPONSE: Thank you for your suggestion. Tables 2 and 3 show the median of the total and partial scores of the SGRQ and SCAQ, not necessarily the sum of the partial scores. Accordingly, it is now indicated in the table that it is not the sum of the partial scores, because of the median of the SGRQ and SCAQ. 

2. Lines 252-268 are just a description of Table 2 and could be shortened,

RESPONSE: Thank you for your suggestion. The content has been condensed per your recommendation.

3. Lines 120-122: "Research assistants ... numbers". This sentence is unnecessarily repeated on line 127.

RESPONSE: Thank you for your observation, with which we agree. Accordingly, the redundant content has been removed. 

4. Data are not provided. Data should be provided as a supplementary file or made available in a public repository. 

RESPONSE: Thank you for your suggestion. To improve clarity regarding the primary outcome, a supplemental file of the data has been provided for your review.

---

## [Decision Letter · Decision Letter 1]

1 Dec 2022

PONE-D-22-13518R1Effectiveness of a telenursing intervention program in reducing exacerbations in patients with chronic respiratory failure receiving noninvasive positive pressure ventilation: A randomized controlled trialPLOS ONE

Dear Dr. Shimoyama,

Thank you for submitting your manuscript to PLOS ONE. After careful consideration, we feel that it has merit but does not fully meet PLOS ONE’s publication criteria as it currently stands. Therefore, we invite you to submit a revised version of the manuscript that addresses the points raised during the review process. Please review the concerns raised in the comments below, in particular the queries re the statistical analysis, sample size power calculations, and references. 

We look forward to receiving your revised manuscript.

Kind regards,

Kathleen Finlayson

Academic Editor

PLOS ONE

Additional Editor Comments:

Thank you for submitting your revised manuscript and responding to the previous reviewer's feedback. On review, I believe there are still some important questions to address in the manuscript.

In particular, the previous query on the need to allow for the longitudinal nature of the data suggests the need for a two-way data analysis, e.g. a Friedman test? I note in table 3 that there is no Mann-Whitney U test result/SE reported, which would be of interest. Another important omission is information on a sample size /power calculation - there are very few outcome events in your study, and as such, it appears underpowered. Providing this information would give a balanced view of your results.

On a minor note, giving the effect size results in the abstract besides wording which is talking about differences in outcomes is a little confusing, I'd just leave in the p value and note which test was done. Avoid including your findings in the introduction/aim section. Further information on the number of patients who fitted the inclusions criteria at recruitment sites, compared to the number who agreed to participate is needed - were all patients who fitted the criteria invited?

Many of the references are very old - updating is needed. As this is an international journal, only references which are available to an international audience should be included. National policy documents should thus have a URL or DOI.

Reviewers' comments:

Reviewer's Responses to Questions

**Comments to the Author**

1. If the authors have adequately addressed your comments raised in a previous round of review and you feel that this manuscript is now acceptable for publication, you may indicate that here to bypass the “Comments to the Author” section, enter your conflict of interest statement in the “Confidential to Editor” section, and submit your "Accept" recommendation.

Reviewer #1: All comments have been addressed

2. Is the manuscript technically sound, and do the data support the conclusions?

Reviewer #1: (No Response)

3. Has the statistical analysis been performed appropriately and rigorously? 

Reviewer #1: (No Response)

4. Have the authors made all data underlying the findings in their manuscript fully available?

Reviewer #1: (No Response)

5. Is the manuscript presented in an intelligible fashion and written in standard English?

Reviewer #1: (No Response)

6. Review Comments to the Author

Reviewer #1: (No Response)

7. PLOS authors have the option to publish the peer review history of their article (what does this mean?). If published, this will include your full peer review and any attached files.

Reviewer #1: No

---

## [Author Response · Author response to Decision Letter 1]

6 Feb 2023

February 4, 2023

Kathleen Finlayson

Academic Editor

PLOS ONE

Thank you for inviting us to submit a revised draft of our manuscript entitled, “Effectiveness of a telenursing intervention program in reducing exacerbations in patients with chronic respiratory failure receiving noninvasive positive pressure ventilation: A randomized controlled trial” to PLOS ONE. The manuscript ID is PONE-D-22-13518R1. We also appreciate the time and effort you and each reviewer have dedicated to providing insightful feedback on ways to strengthen our paper. Thus, it is with great pleasure that we resubmit our article for further consideration. We have incorporated changes that reflect the detailed suggestions you have graciously provided. We also hope that our edits and the responses we provided below satisfactorily address all the issues and concerns you and the reviewers have noted.

To facilitate your review of our revisions, the following is a point-by-point response to the questions and comments delivered in your letter dated 2022.12.2. We hope that the revised manuscript is now suitable for publication in your journal.

Thank you for your consideration. We look forward to hearing from you.

Sincerely,

Makoto Shimoyama,

School of Nursing,

Miyagi University,

1-1 Gakuen, Taiwa-cho, Kurokawa-gun, Miyagi, 981-3298, Japan.

Telephone: +81 22 377 8277

Additional Editor Comments;

1. In particular, the previous query on the need to allow for the longitudinal nature of the data suggests the need for a two-way data analysis, e.g. a Friedman test?

RESPONSE: We would like to thank the reviewer for this comment. In this analysis, the Mann–Whitney U test was used to calculate the corresponding amount of change that occurred before and after the intervention with the same participants and to compare the amount of change between the two groups (intervention and control). As there was no correspondence between the two groups, we do not think there is any problem with the method of analysis using the Mann–Whitney U test. As the reviewer suggested, the Friedman test is used for pre- and post-intervention comparisons; thus, we did not use it in this analysis.

2. I note in table 3 that there is no Mann-Whitney U test result/SE reported, which would be of interest. 

RESPONSE: We would like to thank the reviewer for this comment. The Mann–Whitney U test is a nonparametric test. Thus, we used median and quartile deviations to denote it. We did not add the mean and standard errors because we believe they are used when parametric tests are used.

3. Another important omission is information on a sample size /power calculation - there are very few outcome events in your study, and as such, it appears underpowered. Providing this information would give a balanced view of your results.

RESPONSE: We would like to thank the reviewer for this comment. We did not make a statement regarding sample size/detection power, so we figured it was not clear. We have added a note regarding sample size/power calculations in the “Participants” subsection as follows:

“The average number of hospitalizations among NPPV practitioners in Japan was 2.0 times/year, and it was assumed that the average number of hospitalizations would improve by 1.0 time/year. The required sample size was calculated with a significance level of 5%, a power of 80%, and a standard deviation of the assumed average number of hospitalizations of 1.0 time/year. As a result, the intervention and control groups were set at 15 participants each.” (Lines 95–99)

4. On a minor note, giving the effect size results in the abstract besides wording which is talking about differences in outcomes is a little confusing, I’d just leave in the p value and note which test was done.

RESPONSE: We would like to thank the reviewer for this comment. We agree with the reviewer. The abstract was difficult to understand. We have excluded the effect size and only indicated the p-values. Moreover, we have stated that we used the Mann–Whitney U test for our analysis (Line 34).

5. Avoid including your findings in the introduction/aim section.

RESPONSE: We would like to thank the reviewer for this comment. We have reviewed the text and removed the discussion on the findings of our study that were included in the Introduction. Please check it. 

6. Further information on the number of patients who fitted the inclusions criteria at recruitment sites, compared to the number who agreed to participate is needed - were all patients who fitted the criteria invited?

RESPONSE: We would like to thank the reviewer for this question. The number of patients who met the inclusion criteria at the recruiting facility was 33. We did not explain it well enough. We have added the following part to present this information clearer:

“In total, 33 patients met the participation criteria at the recruiting facility, all of whom were referred by the facility. Of the 33 participants referred by the facilities, 31 (n=15, intervention group; n=16, control group) agreed to participate in the study.” (Lines 198–200)

7. Many of the references are very old - updating is needed. As this is an international journal, only references which are available to an international audience should be included. National policy documents should thus have a URL or DOI.

RESPONSE: We would like to thank the reviewer for this comment. We have reviewed and updated our citations and references. We have also included the URL or DOI as necessary. Please review the references.

---

## [Decision Letter · Decision Letter 2]

2 Jun 2023

PONE-D-22-13518R2Effectiveness of a telenursing intervention program in reducing exacerbations in patients with chronic respiratory failure receiving noninvasive positive pressure ventilation: A randomized controlled trialPLOS ONE

Dear Dr. Shimoyama,

Thank you for submitting your manuscript to PLOS ONE. After careful consideration, we feel that it has merit but does not fully meet PLOS ONE’s publication criteria as it currently stands. Therefore, we invite you to submit a revised version of the manuscript that addresses the points raised during the review process.

Thank you for addressing the previous reviewer's feedback. Please consider the minor recommendations from the current reviewers' comments, to provide extra detail on your methods. 

We look forward to receiving your revised manuscript.

Kind regards,

Kathleen Finlayson

Academic Editor

PLOS ONE

Journal Requirements:

Additional Editor Comments:

Thank you for addressing the previous reviewer's feedback. Please consider the minor recommendations from the current reviewers' comments, to provide extra detail on your methods.

Reviewers' comments:

Reviewer's Responses to Questions

**Comments to the Author**

1. If the authors have adequately addressed your comments raised in a previous round of review and you feel that this manuscript is now acceptable for publication, you may indicate that here to bypass the “Comments to the Author” section, enter your conflict of interest statement in the “Confidential to Editor” section, and submit your "Accept" recommendation.

Reviewer #2: All comments have been addressed

Reviewer #3: (No Response)

2. Is the manuscript technically sound, and do the data support the conclusions?

Reviewer #2: Yes

Reviewer #3: Yes

3. Has the statistical analysis been performed appropriately and rigorously? 

Reviewer #2: Yes

Reviewer #3: Yes

4. Have the authors made all data underlying the findings in their manuscript fully available?

Reviewer #2: Yes

Reviewer #3: Yes

5. Is the manuscript presented in an intelligible fashion and written in standard English?

Reviewer #2: Yes

Reviewer #3: Yes

6. Review Comments to the Author

Reviewer #2: This is a well-designed pragmatic randomized controlled trial that wanted to assess the telenuring intervention program for chronic respiratory failure receiving NPPV. However, some important issues were not clearly reported or conflicting.

1. How long does the telenuring intervention program last in the trial?

2. Considering the sample size estimation, the authors assumed that the average number of hospitalizations would improve by 1.0 time per year. However, the primary outcome of the trial was the number of acute exacerbations in the past 3 months. It is clear that the sample size calculation is not supported by the expected data as the number of exacerbations in the past 3 months is likely less than within one year.

3. As for the statistical analysis, the ITT analysis principle, missing data processing, and co-variate and adjustment methods would be considered.

4. Tests of baseline differences in the table 1 are unnecessary and illogical. Such hypothesis testing is superfluous and can mislead investigators and their readers. In addition, I would just leave in the statics and p value and note which test was done in the table 2 and 3.

Reviewer #3: This appears to be a well thought out and important study to address the needs of NPPV patients that require self-care support. The study is clear and logical and follows the CONSORT guidelines for reporting RCTs. The only comments that I have are minor.

Page 7 Line 114 – Can the investigators please include a short description of the COMPAS system with particular reference to whether this system has an alert function that is sent to clinicians when signs of an acute exacerbation are detected. Please describe the responsibilities of the clinicians in the event that they are not notified. This is because in the Discussion on Page 20 Lines 285-287 ‘Some participants thought that they should not attempt outpatient visits outside of their regular appointments and chose to wait until the next regular outpatient visit, despite having symptoms of an acute exacerbation.’

Page 9 Line 146 – Can the investigators please describe where the health information on the ‘server … managed by a system production and development company’ was stored and did this become part of the patient medical record managed by the hospital. Please also describe the implications of this in the discussion section.

Page 9 Line 152 – Can the investigators please include a description of how the NFC function ‘was used to collect vital signs data’ and how it ‘assisted the participants with inputting data on vital signs.’

Page 10 Lines 165-167 – Can the investigators please include a description of what the SGRO scores represented and likewise for the SCAQ and QOL5 scores so that interpretation of data in the results section is facilitated.

Page 21 Line 301 – can the investigator please clarify what is meant by “clear respiratory symptoms” and what type of “sensations” are being referred to in the sentence:

“Signs of an acute exacerbation include not only the appearance of clear respiratory symptoms but also sensations that are somehow different from the norm”

7. PLOS authors have the option to publish the peer review history of their article (what does this mean?). If published, this will include your full peer review and any attached files.

Reviewer #2: **Yes: **Zehuai Wen

Reviewer #3: No

---

## [Author Response · Author response to Decision Letter 2]

31 Aug 2023

Point by point response to reviewer comments has been uploaded seperately as "Response to Reviewers" file.

---

## [Editor Report · Decision Letter 3]

21 Sep 2023

Effectiveness of a telenursing intervention program in reducing exacerbations in patients with chronic respiratory failure receiving noninvasive positive pressure ventilation: A randomized controlled trial

PONE-D-22-13518R3

Dear Dr. Shimoyama,

We’re pleased to inform you that your manuscript has been judged scientifically suitable for publication and will be formally accepted for publication once it meets all outstanding technical requirements.

Kind regards,

Kathleen Finlayson

Academic Editor

PLOS ONE

Additional Editor Comments (optional):

Thank you for your responses to the reviewers' feedback. Your responses have clarified most of the queries, my one request is in regards to the reviewer's comment: " the ITT principle, missing data processing, and covariate and adjustment methods would be considered". Your response, which was adding a sentence saying they were taken into account, does not give the reader the required information on exactly how you allowed for or processed missing data (or whether there were any at all), or how you allowed for adjusted for covariates. In fact, as you did not undertake multivariable analysis, I suspect that you did not adjust for covariates. This may be appropriate for your study design, however it would be useful to justify this to the readers or make a note of the issue in your study limitations.
---

## [Editor Report · Acceptance letter]

19 Oct 2023

PONE-D-22-13518R3 

Effectiveness of a telenursing intervention program in reducing exacerbations in patients with chronic respiratory failure receiving noninvasive positive pressure ventilation: A randomized controlled trial 

Dear Dr. Shimoyama:

I'm pleased to inform you that your manuscript has been deemed suitable for publication in PLOS ONE. Congratulations! Your manuscript is now with our production department. 

Kind regards, 

on behalf of

Dr. Kathleen Finlayson 

Academic Editor

PLOS ONE